# Control of Mesenchymal Stromal Cell Senescence by Tryptophan Metabolites

**DOI:** 10.3390/ijms22020697

**Published:** 2021-01-12

**Authors:** Kenneth K. Wu

**Affiliations:** 1Institute of Cellular and System Medicine, National Health Research Institutes, 35 Keyan Road, Zhunan Town, Miaoli County 35053, Taiwan; kkgo@nhri.org.tw; Tel.: +886-37-246-166 (ext. 37501); Fax: +886-37-587-408; 2Institute of Biotechnology, College of Life Science, National Tsing-Hua University, Hsinchu 30071, Taiwan

**Keywords:** type 2 diabetes, hyperglycemia, mesenchymal stromal/stem cells, 5-methoxytryptophan, melatonin, cellular senescence mitochondrial dysfunction, reactive oxygen species, antioxidant enzymes

## Abstract

Cellular senescence contributes to aging and age-related disorders. High glucose (HG) induces mesenchymal stromal/stem cell (MSC) senescence, which hampers cell expansion and impairs MSC function. Intracellular HG triggers metabolic shift from aerobic glycolysis to oxidative phosphorylation, resulting in reactive oxygen species (ROS) overproduction. It causes mitochondrial dysfunction and morphological changes. Tryptophan metabolites such as 5-methoxytryptophan (5-MTP) and melatonin attenuate HG-induced MSC senescence by protecting mitochondrial integrity and function and reducing ROS generation. They upregulate the expression of antioxidant enzymes. Both metabolites inhibit stress-induced MSC senescence by blocking p38 MAPK signaling pathway, NF-κB, and p300 histone acetyltransferase activity. Furthermore, melatonin upregulates SIRT-1, which reduces NF-κB activity by de-acetylation of NF-κB subunits. Melatonin and 5-MTP are a new class of metabolites protecting MSCs against replicative and stress-induced cellular senescence. They provide new strategies to improve the efficiency of MSC-based therapy for diverse human diseases.

## 1. Introduction

Cellular senescence is a hallmark of aging [1]. Accumulation of senescent cells promotes aging and triggers age-related disorders [2]. Cellular senescence was originally observed in cultured fibroblasts following limited replications [3]. It was subsequently noted as a response to DNA damage, telomere attrition, mitochondrial dysfunction, and oncogenic, hyperglycemic, and oxidative stresses [4,5,6]. Cellular senescence plays an important role in parturition and embryo development [7,8]. It may influence the fate of tumorigenesis through senescence-associated secretory phenotype (SASP). It was reported that acute senescent cells induce immortalized prostate cells to undergo senescence via SASP but have no effect on metastatic prostate cancer cells [9]. Replicative, developmental, and stress-induced premature cell senescence share common cellular phenotypic changes including an increased expression of p16 and p21, cell cycle and proliferation arrest, senescence-associated (SA) heterochromatin foci, SA-β galactosidase (β gal), and SASP, as well as cellular morphological changes [10]. Phenotypic changes of senescent cells are mediated by multiple signaling pathways leading to complex transcriptional reprogramming [11,12].

Hyperglycemia due to type 2 diabetes (T2D) and pre-diabetic metabolic syndrome and obesity has emerged as a key extracellular stress signal to induce cellular senescence as well as cell death [13]. As T2D is increasing with aging and contributes to age-related chronic diseases [13], hyperglycemia has become a leading age stress factor. Hyperglycemia induces cellular senescence through metabolism shift, reactive oxygen species (ROS) generation, mitochondrial dysfunction, and aberrant gene expressions.

Replicative and stress (hyperglycemia and oxidative stress)-induced mesenchymal stromal cell senescence has been extensively investigated as it is critical for MSC-based cell therapy. Mesenchymal stromal cells (MSCs) are isolated and characterized according to a set of criteria [14,15]. Current isolation procedures generate heterogeneous nonclonal stromal cell populations with different multipotent and differentiation potentials [16]. MSCs possess immunosuppressive and anti-inflammatory properties [17,18]. As MSCs can be obtained and cultured with ease, they are popular sources for cell-based therapy of a variety of human diseases. More than 700 clinical trials have been registered [16]. However, MSC-based cell therapy faces challenging problems. Replicative senescence of cultured MSCs limits the cell expansion and its availability for cell therapy. Moreover, stress-induced premature senescence in vitro and in vivo reduces the efficacy of transplanted MSCs in tissue regeneration and treatment of autoimmune and inflammatory diseases. New strategies are actively being employed to develop new drugs to combat cellular senescence. The candidate drugs are either senolytic, which kill and remove senescent cells, or senomorphic, which modify senescent cell phenotypes to attenuate their tissue-damaging effects [19,20,21]. Senomorphic agents comprise a wide range of compounds with different targets aiming at reducing SASP and senescent markers without causing cell apoptosis [20]. Recent studies indicate that tryptophan metabolites produced via the tryptophan hydroxylase (TPH) pathway defend against replicative and hyperglycemia or oxidative stress-induced cell senescence. 5-methoxytryptophan (5-MTP) was reported to rescue bone marrow mesenchymal stromal cells (BM-MSCs) from high glucose (HG)-induced senescence [22], while melatonin protects MSC from replicative and stress-induced senescence [23]. Melatonin and 5-MTP represent a new class of senomorphic compounds which may be useful in protecting MSC against senescence and age-related diseases. This review will comment on the anti-senescence actions of 5-MTP and melatonin with a focus on 5-MTP biosynthesis, its defense of HG-induced MSC senescence, and mechanisms of actions.

## 2. Hyperglycemia Induces Cellular Senescence

High blood glucose levels (hyperglycemia) contribute to diabetic microvascular, renal, retinal and neural complications by multiple mechanisms including mitochondrial dysfunction and ROS generation [24,25]. Results from in vivo and in vitro experiments have shown that hyperglycemia induces cellular damage, apoptosis, and necrosis through ROS generation and mitochondrial dysfunction [24,25]. In addition, hyperglycemia was reported to induce renal tubular cell and retinal endothelial cell senescence in streptozotocin-induced diabetic mice [26,27] and HG in cultured media was reported to induce senescence of diverse cell types including MSCs [28,29,30]. Senescent cells cause further tissue damage through secretion of pro-inflammatory cytokines and proteolytic enzymes [1,31].

### 2.1. HG-Induced Cellular Senescence Is Attributed to Mitochondrial Dysfunction and ROS Generation

The exact mechanisms by which HG induces senescence are not entirely clear. Mitochondrial dysfunction and ROS generation are considered to be major players. HG induces mitochondrial ROS generation by enhancing mitochondrial metabolism via tricarboxylic acid (TCA) cycle and oxidative phosphorylation [24]. ROS generation is closely related to mitochondrial morphological changes. It was reported that HG-treated rat liver cells undergo mitochondrial fission, which was required for ROS generation [32]. ROS, in turn, cause mitochondrial fission [33], creating a vicious cycle (Figure 1). It was also reported that HG increases ROS through activation of NADPH oxidase [34,35], but its relevance to cell senescence is unclear and remains to be investigated. ROS overproduction is considered to be a major cause of cell damage and lethality. However, at sublethal concentrations, H_2_O_2_ induces cellular senescence as a way of protecting cells from ROS-induced death [36,37]. ROS represent a common mediator via which diverse stress signals induce cellular senescence. For example, Ras overexpression in fibroblasts induce cellular senescence by elevation of ROS generation [37].

ROS induces cellular senescence by oxidative damage to DNA, leading to telomere attrition and altered expression of p53, p16, and p21 [38,39]. However, increased ROS generation cannot explain all the phenotypic manifestations. Mitochondrial structural changes and functional defects contribute significantly to stress-induced senescence [40]. Changes in mitochondrial dynamics [41,42], metabolism, and signaling molecules such as AMPK [43,44,45] are considered to mediate senescence independent of ROS.

### 2.2. HG Induces MSC Senescence

MSCs in bone marrow reside in hypoxic microenvironment. They depend on glycolysis as energy source and thus express relatively low levels of oxidative phosphorylation (OXPHOS) proteins [46]. MSCs cultured in nutrient-rich normoxic conditions shifts the metabolism to OXPHOS [47]. MSC metabolism is altered during osteoblastic vs. chondrogenic differentiation. Osteoblastic differentiation requires OXPHOS, while chondrogenic differentiation uses aerobic glycolysis in energy generation [48]. When MSCs are incubated with HG medium, excessive intracellular glucose shifts metabolism from aerobic glycolysis to TCA cycle and OXPHOS. Consequently, a high level of ROS is leaked from the electron transport chain which induces MSC senescence [49,50]. Effects of HG on MSC mitochondrial biogenesis and metabolism have not been described. However, it is likely HG-induced mitochondrial structural and metabolic changes contribute to MSC senescence. MSC senescence puts a limit to MSC expansion, which hampers its use in cell therapy. Furthermore, it impairs MSC function which reduces its support for hematopoiesis and immunosuppressive properties [51].

## 3. 5-MTP Rescues MSCs from HG-Induced Senescence

### 3.1. 5-MTP Biosynthesis and Its Perturbation by Environmental Stresses

5-MTP was originally identified as a cytoprotective molecule named cytoguardin [52]. It is produced in and released from human fibroblasts [52]. Its biosynthesis in fibroblasts is catalyzed by two enzymes: tryptophan hydroxylase (TPH), which converts L-tryptophan to 5-hydroxytryptophan (5-HTP) and hydroxyindole O-methyltransferase (HIOMT), which converts 5-HTP to 5-MTP [53] (Figure 2A). Of the two isoforms of TPH identified and characterized in human cells [54,55], TPH-1 is selectively expressed in human fibroblasts, and silencing of TPH-1 with siRNA results in diminished release of 5-MTP into the cultured medium [53]. Thus, TPH-1 is the functional isoform catalyzing 5-MTP synthesis. HIOMT was previously identified and characterized as the terminal enzyme in catalyzing melatonin (N-acetyl-5-methoxytryptamine) synthesis. As it catalyzes the conversion of N-acetylserotonin to melatonin in pineal cells (Figure 2B), it is commonly called N-acetylserotonin O-methyltransferase (ASMT). It is encoded by a single gene with three mRNA isoforms due to alternative splicing [56,57]. The full-length isoform which contains LINE 1 repeat sequences in exon 6 codes for a 373 aa protein [57]. The isoform that codes for a 345 aa protein has exon 6 spliced, while the isoform coding for a 298 aa protein loses exons 6 and 7 to splicing. The nomenclature for HIOMT isoforms is different between NCBI and Uniprot database. To avoid confusion, the isoforms are thus named HIOMT373, 345 and 298, respectively [58]. Bovine and macaque express only a single transcript which aligns with human HIOMT345. As only HIOMT345 from pineal tissues is catalytically active in melatonin synthesis, HIOMT345 is considered to be a wild-type ASMT [59]. By contrast, HIOMT345 is not involved in 5-MTP production. Fibroblasts express only HIOMT298 isoform which was shown to be active in catalyzing 5-MTP synthesis [58]. This is surprising because HIOMT298 is a truncated isoform and structural analysis suggests that it lacks binding site for S-adenosylmethionine, a co-factor required for melatonin synthesis [59]. It is unclear how this truncated isoform catalyzes conversion of 5-HTP to 5-MTP.

It was subsequently reported that 5-MTP production is not limited to fibroblasts. Vascular endothelial cells (ECs) and smooth muscle cells (SMCs) as well as bronchial and renal epithelial cells produce 5-MTP [60]. Human umbilical vein ECs express TPH-1 and HIOMT298. Immunofluorescent studies show that 5-MTP is detected in cytoplasm with an endoplasmic reticulum (ER) pattern, and biochemical studies suggest that 5-MTP is secreted via Golgi vesicular transport [60]. BM-MSCs, like fibroblasts and ECs, express TPH-1 and HIOMT298 and release 5-MTP into the conditioned medium.

Lipopolysaccharide (LPS) and pro-inflammatory cytokines inhibit 5-MTP production by suppressing TPH-1 expression in ECs [60,61]. Addition of 5-MTP alleviates LPS and cytokine-induced vascular permeability suggesting that 5-MTP plays an important role in protecting endothelial barrier function [60,61]. LPS and pro-inflammatory cytokines exert their actions via ROS [62,63]. BM-MSCs cultured in medium containing HG release a lower amount of 5-MTP into the medium than control. Incubation of BM-MSCs with sublethal concentrations of H_2_O_2_ results in reduction of 5-MTP in the medium. Reduction of 5-MTP may be due to TPH-1 suppression by ROS.

### 3.2. 5-MTP Rescues BM-MSC from HG-Induced Cellular Senescence by Suppressing ROS Generation

Addition of 5-MTP to BM-MSC cultured in HG medium prevents growth arrest, blocks p16 and p21 elevation, attenuates SA-β gal positive cells and reduces interleukin-6 (IL-6) [22]. Furthermore, 5-MTP preserves BM-MSC morphology. 5-MTP controls HG-induced MSC senescence by suppressing ROS accumulation [22]. HG enhances mitochondrial ROS generation through electron transport chain [24]. Excessive ROS causes cell damage and death. However, at sublethal concentrations, ROS induces cellular senescence. Pretreatment of BM-MSC with 5-MTP results in reduction of ROS [22]. ROS comprise several species of oxidants and oxygen radicals among which H_2_O_2_ is a key mediator of cellular senescence. H_2_O_2_ at sublethal concentrations induces characteristic BM-MSC senescence [22]. 5-MTP pretreatment alleviates H_2_O_2_-induced cellular senescence [22].

### 3.3. 5-MTP Upregulates MnSOD and Catalase via FOXO3a

ROS levels in mitochondria are controlled by several antioxidant enzymes including MnSOD (SOD-2) and catalase [64]. Hyperglycemia was reported to alter MnSOD and catalase activities in a cell-dependent manner: it downregulates MnSOD and catalase in human umbilical vein ECs but not in microvascular EC [65]. HG does not alter MnSOD or catalase activity in BM-MSC [22]. However, 5-MTP upregulates MnSOD and catalase in HG-treated BM-MSCs. MnSOD and catalase are mitochondrial antioxidant enzymes that have immediate access to ROS generated in mitochondrial matrix, converting superoxide to water and oxygen. Upregulation of both enzymes by 5-MTP increases ROS scavenging and thereby reduces the damaging effect of ROS.

FOXO3a is a pleotropic transcriptional activator that mediates expression of diverse genes including antioxidant genes [66,67]. 5-MTP upregulates FOXO3a expression, and silencing of FOXO3a in HG-treated BM-MSC abrogates 5-MTP-induced rise of MnSOD and catalase [22]. Importantly, FOXO3a silencing eliminates the protective effect of 5-MTP on HG-induced senescence [22]. 5-MTP protects HG-induced cellular senescence by upregulating FOXO3a-mediated MnSOD and catalase expression.

### 3.4. 5-MTP Restores BM-MSC Osteogenic Differentiation via Controlling ROS Levels

Excessive ROS generation was reported to induce defective MSC differentiation: it impairs osteogenic and promotes adipogenic differentiation [68]. The adverse effects of ROS on MSC differentiation play an important role in age-related skeletal disorders and obesity. It is worth noting that physiological osteogenic differentiation is accompanied by metabolism shift from aerobic glycolysis to oxidative phosphorylation [46]. Although ROS is expected to be elevated, it is not because MnSOD and catalase activities are upregulated [46]. Thus, the ROS level is controlled by a well-regulated redox balance during osteogenic differentiation. Oxidative stress disrupts the balance and tilts it toward ROS overproduction, which impairs osteogenic differentiation. 5-MTP restores the redox balance and thereby protects MSC osteogenic differentiation [22]. It is less clear whether excessive ROS influence MSC chondrocyte differentiation [69]. However, it was reported that chondrocytes from patients with osteoarthritis (OA) exhibit mitochondrial defects and senescent changes [70]. Oxidative stress-induced chondrocyte senescence contributes to joint dysfunction in OA [70]. As 5-MTP and melatonin reduce ROS accumulation, they may exert an effect on protecting chondrocyte function [22,71].

## 4. Melatonin Protects Against Replicative and Stress-Induced Cellular Senescence

Melatonin is produced primarily in pineal and retinal cells. Its synthesis from L-tryptophan shares with serotonin biosynthesis common enzymes, i.e., TPH-2 which converts L-tryptophan to 5-HTP and aromatic amino acid decarboxylase (AADC) which catalyzes decarboxylation of 5-HTP to form 5-hydroxytryptamine (5-HT, serotonin) (Figure 2B). 5-HT is N-acetylated by arylalkylamine N-acetyltransferase (AANAT) to form N-acetyl-5HT (N-acetylserotonin). AANAT expression is regulated by circadian rhysm and its expression in dark accounts for burst melatonin synthesis. The final step of melatonin synthesis is catalyzed by HIOMT (ASMT). All three isoforms are detected in pineal cells but only isoform 345 is catalytically active in melatonin synthesis [59]. As HIOMT298 is expressed in pineal cells, it is possible that they may produce 5-MTP. Melatonin plays a physiological role in regulating circadian rhythm and sleep [72]. In addition, a large number of reports suggest that melatonin possess anti-inflammatory actions by suppressing the expression of pro-inflammatory genes such as cyclooxygenase-2 [73]. It protects against stress-induced cell and tissue damage [74]. Melatonin inhibits cancer growth through melatonin-receptor-mediated signaling transduction [75,76]. Recent studies reveal that melatonin also possesses anti-senescence activities.

### 4.1. Melatonin Attenuates MSC Replicative Senescence by Restoring Mitochondrial Function and Reducing ROS

Cultured MSCs undergo progressive replicative senescence accompanied by mitochondrial dynamic changes, excessive ROS generation, and decreased mitochondrial membrane potential [23]. Melatonin pretreatment rescues MSC from replicative senescence by restoring mitochondrial dynamics, reducing ROS generation and maintaining membrane potential through upregulating heat shock protein 1L (HSPA1L) [18]. HSPA1L, a member of HSP70 family [77], functions as a chaperone protein facilitating protein folding and stabilizing prion protein Prp^c^ [23]. Prp^c^ regulates mitochondrial integrity and function through binding to HSPA1L. Melatonin-induced HSPA1L upregulation results in recruitment of Prp^c^ to mitochondria to maintain mitochondrial integrity. HSPA1L expression is suppressed in replicative senescent MSCs with a lower level of Prp^c^ recruitment [23]. Melatonin pretreatment restores HSPA1L expression and Prp^c^ recruitment, thereby alleviating senescent changes and improving MSC functions. Melatonin-treated MSCs confer more effective revascularization when transplanted to a hindlimb ischemic murine model [23].

### 4.2. Melatonin Controls HG- and Oxidant-Induced Cellular Senescence by Upregulating Antioxidant Enzymes

Pancreatic β-cells incubated in HG medium undergo premature senescence which is associated with reduced expression of antioxidant enzymes and increased ROS [78]. HG-treated cells exhibit impaired insulin secretion. Melatonin attenuates HG-induced senescence and improves insulin secretory activity by restoring expression of antioxidant enzymes and suppressing ROS generation [78]. Melatonin was reported to protect MSC from oxidant-stress-induced senescence by scavenging ROS generation [79]. Melatonin is also effective in antagonizing the action of a uremic toxin, p-cresol, on MSC senescence [80]. Melatonin blocks HG- and oxidant-induced senescence in diverse cell types by scavenging ROS generation through upregulation of MnSOD and catalase activities [78,80] in a manner analogous to 5-MTP.

## 5. Melatonin and 5-MTP Target p38 MAPK

p38 MAPK occupies a central position in transducing signals from environmental insults [81]. It mediates inflammation by activating pro-inflammatory activators [81]. Furthermore, it mediates stress-induced cellular senescence through interaction with ROS [82,83]. ROS alters p38 MAPK activities by oxidative modification of kinases in the p38 MAPK signaling cascade [84]. ROS may sustain p38 MAPK activation by inhibiting MAPK phosphatases (MKP) [85,86].

5-MTP was reported to be an arsenal against LPS- induced systemic inflammation by blocking p38 MAPK activation [60]. It protects vascular barrier function and endothelial integrity by inhibiting p38-mediated damage [61,87,88]. 5-MTP effectively prevents vascular intimal hyperplasia and vascular smooth muscle cell migration by blocking p38 MAPK activity [89]. Thus, p38 MAPK inactivation is a key mechanism by which 5-MTP exerts its biological actions. It is highly likely that 5-MTP attenuates HG-induced MSC senescence by suppressing p38 MAPK signaling pathway. Melatonin controls inflammation and tumorigenesis also by targeting p38 MAPK. Melatonin was reported to inhibit breast cancer cell invasion and control glial cell-mediated inflammation by blocking p38 MAPK activation [90,91]. Furthermore, it protects BM-MSC from oxidant-induced senescence and differentiation defects through inhibiting p38 MAPK [92].

5-MTP and melatonin protect stress-induced cell damage and senescence by targeting ROS and p38 MAPK-signaling pathway suggesting that they share a common mechanism. Melatonin is known to exert its actions by interacting with membrane receptors. It is unknown whether 5-MTP acts via a specific receptor. There is suggestive evidence that macrophage plasma membrane expresses 5-MTP receptors [60]. It remains unclear whether 5-MTP receptors are related to melatonin receptors and whether the receptor-mediated signaling pathway cross-talks with the p38 MAPK signaling cascade.

## 6. Melatonin and 5-MTP Control Cellular Senescence through Inhibition of NF-κB

Cellular senescence is accompanied by transcriptome rearrangement and altered gene expressions [11,12]. Several transactivators are activated to promote SASP and senescence. Genome-wide search and proteomic analysis identify NF-κB as a master regulator of cellular senescence and SASP [93,94]. ROS and redox transitions are the key drivers of NF-κB activation [95,96,97]. H_2_O_2_ activates IκB kinase (IKK) via which it phosphorylates and degrades IκB and the consequent liberation of p65/p50 NF-κB [95]. NF-κB enters nucleus where it is phosphorylated and binds to specific binding motifs of a large repertoire of pro-inflammatory and pro-senescent genes and promotes their expression. C/EBPβ was reported to regulate stress-induced senescence and SASP via ROS and p38 MAPK [98]. NF-κB and C/EBPβ act in concert to promote senescence, and mediate age-related chronic inflammation and tissue damages through SASP. Melatonin and 5-MTP block NF-κB and C/EBPβ activation and thereby attenuate stress-induced senescence and SASP-mediated chronic inflammation.

### Melatonin and 5-MTP Modify Histone and NF-κB Acetylation

Gene transcription is regulated by epigenetic modification of histones and transactivators. Acetylation of histones by transcriptional co-activators such as p300 histone acetyltransferase (HAT) enhances promoter activities by altering chromatin structure [99,100]. Furthermore, p300 HAT acetylates myriade transactivators to strengthen their binding [99,100]. Histone and transactivator acetylation are controlled by histone deacetylases and a dynamic balance between p300 HAT and deacetylases maintains a normal state of gene expression. NF-κB binding and transcriptional activity are enhanced by p300 HAT [101,102]. HG increases expression of p300 [103]. 5-MTP inhibits p300 HAT activation and reduces NF-κB-mediated expression of cyclooxygenase-2 (COX-2) and pro-inflammatory cytokines [60,104]. 5-MTP probably controls stress-induced MSC senescence by blocking p300 HAT activation thereby reducing NF-κB acetylation and NF-κB mediated transcription of pro-senescent genes.

Melatonin protects cells and tissues from inflammatory damage by inhibiting NF-κB activation and NF-κB-mediated expression of COX-2 and cytokines [105,106,107]. Furthermore, its control of H_2_O_2_-induced cellular senescence is mediated by Sirt-1 dependent deacetylation of p65 submit of NF-κB, thereby reducing NF-κB activity [108]. Sirt-1 is a NAD-dependent deacetylase, which performs diverse biological functions including regulation of cellular senescence [109]. Sirt-1 is suppressed in senescent cells [110] and its supplement alters senescent phenotype and reduces secretion of pro-inflammatory cytokines through histone de-acetylation [111]. H_2_O_2_ at sublethal concentrations suppresses Sirt 1 expression accompanied by p38 MAPK activation. Melatonin restores Sirt 1 levels while inhibiting p38 MAPK suggesting that Sirt 1 is regulated via p38 MAPK and melatonin upregulates Sirt 1 by controlling p38 MAPK.

Transcription of pro-senescent and pro-inflammatory genes is regulated by a delicate balance between p300 HAT and Sirt 1 deacetylase. Oxidative and metabolic stresses tilt the balance to histone and NF-κB acetylation through p300 HAT activation and Sirt 1 downregulation (Figure 3). Melatonin restores the balance and maintains a normal level of histone and NF-κB acetylation. Given that 5-MTP inhibits p300 HAT, 5-MTP acts on histone and NF-κB acetylation in a manner similar to melatonin. This regulatory mechanism is of particular importance to SASP as it contributes significantly to age-related chronic inflammatory disorders.

## 7. Conclusions

Melatonin and 5-MTP are structurally related metabolites derived from L-tryptophan via the TPH pathway (Figure 2C). They control replicative and/or HG- and oxidant-induced MSC senescence by scavenging ROS. Replicative and oxidative stress-induced cellular senescence is characterized by an early event of mitochondrial structural changes and metabolic shift, resulting in excessive ROS generation. Melatonin and 5-MTP reduce ROS by upregulating the expression of mitochondrial MnSOD and catalase. Melatonin protects mitochondrial integrity and function by upregulating a p70 heat shock family protein, HSPA1L, which recruits Prion protein, Prp^c^, to maintain mitochondrial homeostasis and reduce ROS generation.

Proteomic analysis and genome-wide search identified NF-κB (p65/p50) as a master promoter of cellular senescence. Stress signals activate IKK, which phosphorylates IκB, resulting in IκB degradation and p65/p50 activation. NF-κB activity is post-translationally enhanced by phosphorylation and epigenetic modification. The p300 HAT acetylates NF-κB subunits and augments its binding activity, while Sirt 1 deacetylates NF-κB and reduce its activity. Pro-inflammatory mediators and HG increase p300 HAT activity, thereby enhancing NF-κB-mediated senescence. Stress-induced senescence is accompanied by downregulation of Sirt 1, enforcing NF-κB acetylation and activation. 5-MTP inhibits p300 HAT activity, thereby reducing NF-κB binding and transcriptional activity, while melatonin upregulates Sirt 1 and the consequent deacetylation of p65 subunit. Epigenetic modification of NF-κB and histone is a common transcriptional mechanism by which tryptophan metabolites combat cellular senescence.

Several triggering events of cellular senescence are mediated via p38 MAPK. The p38 MAPK mediates ROS-induced cellular senescence and NF-κB activation. Melatonin and 5-MTP target p38 MAPK for their anti-senescent and anti-inflammatory actions. It is unclear how these two metabolites block p38 MAPK activation. Melatonin exerts its actions by interaction with membrane receptors and receptor-mediated signaling transduction. It is likely that melatonin inhibits p38 MAPK and transcriptional mechanism through cross-talk between its receptor-mediated signaling and the p38 MAPK activation cascade. Preliminary data suggest that 5-MTP acts via interaction with a membrane receptor. However, the receptor has not been isolated and characterized and it’s signaling pathway remains to be investigated.

Melatonin-treated MSCs in culture improve revascularization when transplanted to a hindlimb ischemia mouse model through attenuation of cellular senescence. Melatonin and 5-MTP are potentially useful for improving MSC expansion and restoring MSC function in MSC-based cell therapy of diverse age-related disorders.

## Figures and Tables

**Figure 1 ijms-22-00697-f001:**
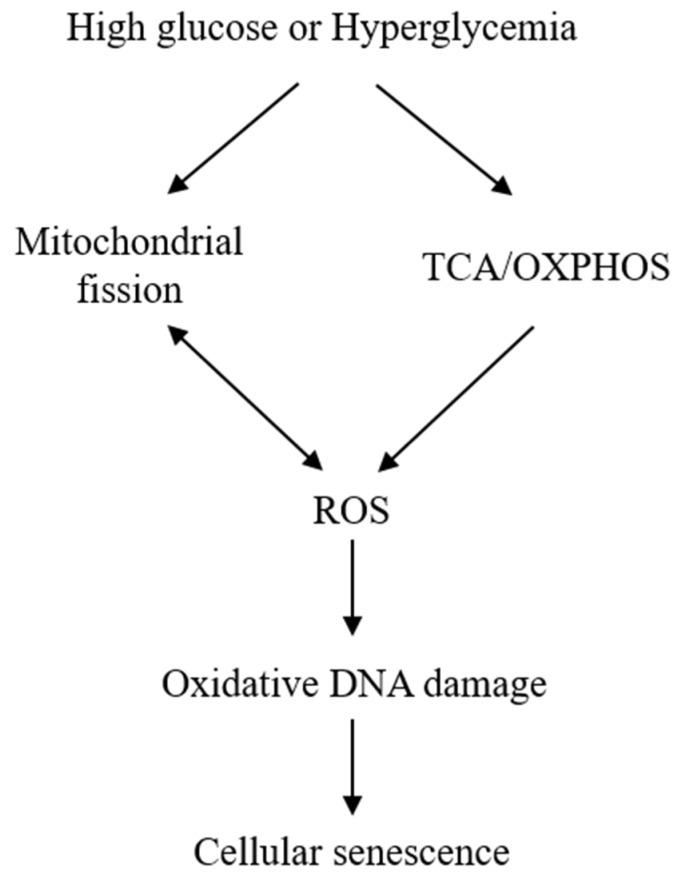
**Simplified scheme illustrating the mechanism by which high glucose (HG) induces cellular senescence**. HG enhances tricarboxylic acid (TCA) cycle and oxidative phosphorylation (OXPHOS), which results in reactive oxygen species (ROS) generation. HG induces mitochondrial fission, which is accompanied by ROS accumulation. ROS, in turn, induces mitochondrial fission as indicated by a two-head arrow. Sublethal ROS (H2O2) induces oxidative DNA damage and senescent phenotypic changes.

**Figure 2 ijms-22-00697-f002:**
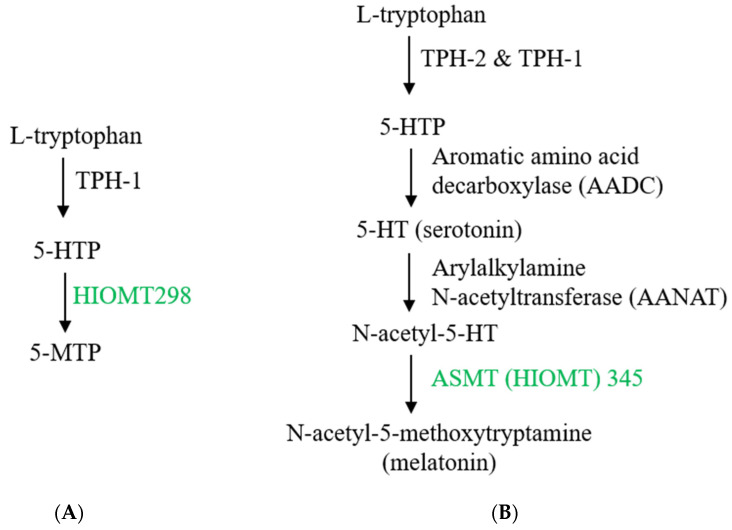
**Biosynthesis of 5-methoxytryptophan (5-MTP) and melatonin.** (**A**) 5-MTP is produced in human fibroblasts, bone marrow mesenchymal stromal cells (BM-MSCs), vascular endothelial cells (ECs) and SMCs, bronchial and renal epithelial cells. Its synthesis is catalyzed by TPH-1 (tryptophan hydroxylase-1) followed by hydroxyindole O-methyltransferase (HIOMT, also known as N-acetylserotonin O-methyltransferase, ASMT). HIOMT298 denotes HIOMT isoform coding for 298 aa HIOMT. (**B**) For comparison, melatonin biosynthesis in pineal cells is shown, highlighting HIOMT345 (ASMT 345) as the functional isoform. (**C**) Structural comparison between 5-MTP and melatonin.

**Figure 3 ijms-22-00697-f003:**
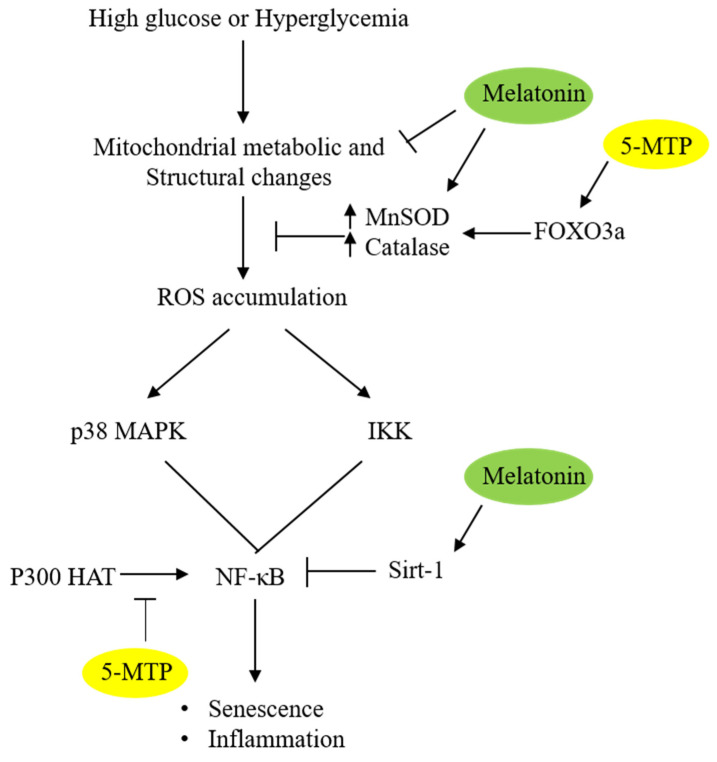
**Schematic illustration of the anti-senescent actions of 5-MTP and melatonin.** 5-MTP upregulates MnSOD and catalase activities via FOXO3a, thereby reducing ROS accumulation. Melatonin reduces ROS by upregulating MnSOD and catalase possibly via maintaining mitochondrial integrity. 5-MTP inhibits p300 HAT thereby reducing NF-κB activity while melatonin upregulates Sirt 1 which de-acetylates p65 and suppresses NF-κB activity.

## Data Availability

Non-applicable.

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
