# Peer review of "Control of Mesenchymal Stromal Cell Senescence by Tryptophan Metabolites"

_ijms, 2021, doi:10.3390/ijms22020697_

Round 1

Reviewer 1 Report

In this review, the author described the role of Tryptophan metabolites such as 5-methoxytryptophan (5-MTP) and melatonin to attenuate HG-induced MSC senescence by protecting mitochondrial integrity and function and reducing ROS generation.

The review is appropriate and well performed, clearly written and with a very updated bibliography.

I have some comments:

  • Author have to report a detailed and proper definition for mesenchymal stromal cells (MSCs) in “Introduction” paragraph. Stromal cells are heterogeneous and contain several populations, including stem cells. The term MSCs should be used for mesenchymal stromal cells rather than mesenchymal stem cells since the cells they refered do not contain a pure population of stem cells. Indeed, the authors should better explain that the isolation of MSCs according to current criteria produces heterogeneous, non-clonal cultures of stromal cells containing stem cells with different multipotential properties, committed progenitors, and differentiated cells. To help in proper definitions please see: Cell Transplant. 2016;25(5):829-48. doi: 0.3727/096368915X689622. PubMed PMID: 26423725.

  • Author could add more information about senescence. The senescent cells have the ability to elicit SASP. Some investigations showed that the SASP released by senescent fibroblasts may promote cancer. Nevertheless, there are other studies evidencing that the secretome of senescent fibroblasts may induce growth arrest and apoptosis of cancer cells (PMID: 31412320). Cellular senescence is also implicated in maintaining proper homeostasis during pregnancy, human parturition (doi:10.1016/j.ajog.2015.05.041), and fetal development (doi:10.1016/j.cell.2013.10.041), suggesting that senescent cells are beneficial during early development.

Author Response

Thank the reviewer for positive comments. Points raised are well taken.

  1. The reviewer raised an important point. Per reviewer’s suggestion, a statement pertaining to MSC heterogeneity is added to the text (p. 4 lines 3-5).
  2. Role of cellular senescence in parturition and embryo development as well as in tumorigenesis has been addressed in the text (p.3 lines 5-8). Relevant references suggested by the reviewer are added (Ref. #7-9).

Reviewer 2 Report

The author of this review article has described the control of cellular senescence in mesenchymal stromal cells by tryptophan metabolites. This articles describes the metabolism of MSCs with respect to aerobic and anaerobic glycolysis and how high glucose concentration (hyperglycemia) induces cellular senescence. The article details the effect of melatonin and how this helps to reduce cellular senescence via inhibition of p38 MAP kinase and NF-kB pathway. The article provides an interesting focus area for research and is a nice review.

The author can further improve the articles, by addressing the following points:

  1. For section 2, a summary figure describing senescence with a specific emphasis on mitochondria and hyperglycemia should be included. This would help to illustrate the points discussed in the preceding section.
  2. The effect of 5-MTP on MSC osteogenic differentiation has been discussed in the article. However, how does excessive ROS generation affect MSC chondrogenesis ? This is an area of research that has been addressed (articles by Buckwalter J, Martin JA) with respect to oxidative stress and it is well known that chondrocytes and MSC chondrogenesis undergo anaerobic glycolysis, as stated in the article. How do tryptophans affect this process and are there any previous literature ? Please add a section on this area.
  3. A new area of research is the use of senolytics and senomorphics. Though the articles is focussed of trytophans, the authors should mention these drug forms, as senomorphics are known to reduce senescence associated secretory proteins (SASPs) and potentially reduce ROS generation. A comment from the author on these drugs with a focus on the points addressed in this review should be included.

Author Response

Reviewer’s comments are appreciated. Points raised are addressed individually as follows.

  1. A figure summarizing section 2 is added (Figure 1).
  2. This point is well taken. Influence of excessive ROS on MSC chondrogenic differentiation has not been extensively characterized. Nor is the effect of tryptophan metabolites on chondrocyte differentiation clearly defined. A statement pertaining to chondrocyte differentiation has been added to the text (p. 11, lines 7-12). Relevant references are added (Ref. #19 & 20).
  3. The reviewer raised an important point. A statement on senolytics and senomorphics is included in the revised manuscript (p. 4, lines 5-9 from the bottom).

Round 2

Reviewer 2 Report

The authors has answered the questions appropriately and addressed my suggestions.

For figure 1, there appear to be no changes to the figure apart from the figure legend. Please amend the figure to follow the new figure legend.

Author Response

A new Figure 1 was added to the revised manuscript. As it may be misplaced, the revised manuscript with three figures including the added Figure 1 is re-submitted.
